# Feline Oral Squamous Cell Carcinoma: A Critical Review of Etiologic Factors

**DOI:** 10.3390/vetsci9100558

**Published:** 2022-10-11

**Authors:** Inês Sequeira, Maria dos Anjos Pires, José Leitão, Joaquim Henriques, Carlos Viegas, João Requicha

**Affiliations:** 1Department of Veterinary Sciences, University of Trás-os-Montes e Alto Douro, 5000-801 Vila Real, Portugal; 2Animal and Veterinary Research Center (CECAV), University of Trás-os-Montes e Alto Douro, 5000-801 Vila Real, Portugal; 3Associate Laboratory for Animal and Veterinary Sciences (AL4AnimalS), 5000-801 Vila Real, Portugal; 4Department of Sport, Exercise and Health Sciences, University of Trás-os-Montes e Alto Douro, 5000-801 Vila Real, Portugal; 5Oncology Service, Centre Hospitalière Vétérinaire Frégis, 94110 Arcueil, France

**Keywords:** cat, oral cavity, squamous cell carcinoma, critical review, etiology, papillomavirus

## Abstract

**Simple Summary:**

Feline oral squamous cell carcinoma (FOSCC) is the most common oral neoplasia in cats. This malignant tumor is locally invasive, has a high mortality rate, and its etiology is not yet known. A critical review about the potential etiologic factors of FOSCC was performed, considering publications between 2000 and 2022. The initial search resulted in 553 publications, with only 26 of these being included in the review. Sixteen studies were related to viral etiology and nine related to environmental factors such as exposure to tobacco smoke, ectoparasitic products, and the presence of oral comorbidities. When evaluated, feline papillomavirus was detected in 16.2% of samples of FOSCC. In the three studies focused on exposure to tobacco smoke, 35.2% (30/85) of cats with FOSCC had a history of this exposure. Among 485 cats with FOSCC, 6.4% had dental and oral pathology. The present study demonstrates that the available evidence on the etiology of FOSCC is still limited, however, there has been increasing interest in this topic.

**Abstract:**

Feline oral squamous cell carcinoma (FOSCC) is the most common oral neoplasia in cats. This malignant tumor is locally invasive, has a high mortality rate, and its etiology is not yet known. In humans, head and neck squamous cell carcinoma is associated with tobacco smoke, alcohol consumption, and human papillomavirus infection. Herein, a critical review about the potential etiologic factors of FOSCC was performed, considering publications between 2000 and 2022, aiming to synthesize all available scientific evidence regarding this issue. Recommendations of the PRISMA statement and the Cochrane Collaboration were followed and the PubMed database searched by using the MeSH terms MeSH terms “oral”, “mouth”, “lingual”, “labial”, “gingiva”, “carcinoma”, “squamous”, and “feline”. The selection process for eligible studies was based on specific inclusion and exclusion criteria and the quality of the studies assessed. The initial search resulted in 553 publications, with only 26 of these being included in the review. Sixteen studies were related to viral etiology and nine related to environmental factors such as exposure to tobacco smoke, ectoparasitic products, and the presence of oral comorbidities. When evaluated, feline papillomavirus was detected in 16.2% of samples of FOSCC. In the three studies focused on exposure to tobacco smoke, 35.2% (30/85) of cats with FOSCC had a history of this exposure. The consumption of canned food and the use of deworming collars were associated, in only one publication, with a risk of neoplasia increased by 4.7 and 5.3 times, respectively. Among 485 cats with FOSCC, 6.4% had dental and oral pathology (i.e., periodontal disease or feline chronic gingivostomatitis). The present study demonstrates that the available evidence on the etiology of FOSCC is still limited, however, there has been an increasing interest on this topic. To better understand the role of the possible etiological factors of this aggressive disease, and model for its human counterpart, large, prospective multi-institutional studies are needed.

## 1. Introduction

Squamous cell carcinoma (SCC) is the most common malignant oral tumor in cats. In general, feline oral squamous cell carcinoma (FOSCC) grows fast and is locally invasive, causing destruction of oral tissues [1,2]. The survival rate of cats affected by oral SCC has not improved over recent decades and the prognosis remains poor independently of the treatment instituted [3,4,5,6,7]. Considering the similarities in incidence, tumor biology, therapy, and prognosis, FOSCC has been proposed as a spontaneous model for human head and neck squamous cell carcinoma (HNSCC) [8,9]. HNSCC is the sixth most common human cancer and risk factors include tobacco, alcohol consumption, chronic inflammation, and human papilloma virus [10].

Despite the intense scientific research on this tumor, its etiology remains unclear. The purpose of this study was to identify etiologic factors for the development of FOSCC and synthesize all available scientific evidence regarding this issue. Knowing risk factors for FOSCC is essential for veterinarian and pet owners to understand and develop prevention strategies to reduce cancer incidence.

## 2. Materials and Methods

This critical review (CR) was conducted in accordance with the Preferred Reporting Items for Systematic Reviews and Meta-Analyses (PRISMA) declaration and followed the recommendations of the Cochrane Collaboration (Cochrane Handbook for Systematic Reviews of Interventions).

### 2.1. Literature Search

The literature search was conducted in June 2020. The database PubMed was searched using the following MeSH entry terms mixed with Boolean phrases ‘AND’ or ‘OR’: Oral OR Mouth OR Lingual OR Labial OR Gingiva AND Carcinoma OR Squamous AND Feline.

### 2.2. Study Selection

Following the elimination of duplicates, abstracts were assessed by two reviewers (IS and JR) to identify and select studies that met the inclusion criteria. Abstracts indicating that the study did not meet the eligibility criteria were excluded at this stage. For the remaining studies, the full texts of studies were assessed by the same two reviewers (IS and JR) for inclusion in the critical review according to the criteria. For those studies selected for full text screening, the bibliographical reference lists were searched for other relevant references.

The following criteria were applied for the selection of eligible studies: (i) studies using/that included oral squamous cell carcinoma samples from domestic cats, (ii) studies that identified possible etiologic agents of feline oral squamous cell carcinoma (FOSCC), including the ones whose main objective was not the etiology, (iii) studies published in English, French, Portuguese, or Spanish, and (iv) analytic observational retrospective or prospective studies and descriptive studies, such as case series and case reports.

Studies were excluded if (i) not published in English, (ii) were dissertations or theses, (iii) included non-oral tumor samples in the analysis, (iv) included tumor samples without mentioning the location or the type of tumor, and (v) the full text was not available online.

### 2.3. Data Extraction

The following information was extracted from the included studies: authors, date of publication, geographic location, study design, number of cases, cats’ breed, life stage, and sex, and information about potential etiological agents (bacterial, fungus, viral infection, environmental factors) and oral comorbidities. The animals’ lifespan was divided into 6 groups: kitten (0–6 months), junior (7 months-2 years), primer (3–6 years), mature (7–10 years), senior (11–14 years), and geriatric (>15 years) [11].

### 2.4. Assessment of Qualified Studies

The quality of the studies was assessed using the methodological criteria proposed by Downs and Black, scored on 27 items of a checklist, which was based on reporting aspects, and external and internal validity. In order to assess the studies, only nine items were considered. The remaining items were excluded as there were not experimental studies with statistical power. The level of evidence of those studies were rated/classified using a scoring system of strongest (I) to weakest (V) evidence modified from a grading system published by the Oxford Centre for Evidence-Based Medicine (Appendix A).

## 3. Results

The PRISMA flow chart for the selected articles used in this systematic review can be seen in Figure 1.

Six hundred and twenty-eight references were obtained in the initial search made on PubMed/MEDLINE’s database, in September of 2022. After title and abstract screening, 454 articles were excluded and 174 articles were selected for final full text review, which referred to (i) cats with oral SCC, including those whose main objective of the study was not the etiology, (ii) cats with oral SCC, but without referencing the specific location, and (iii) cats with oral neoplasia, but the type of tumor was not referenced.

After the full reading of the 174 selected studies, only 26 original studies were included in this critical review. One hundred and forty-eight studies were excluded for the following reasons: (i) no etiological factor of the disease was studied (n = 82), (ii) articles of literature review (n = 14), (iii) studies using cell lines of FOSCC and not related to clinical trials (n = 7), and (iv) no FOSCC biopsy was included or the location of the lesion was not referred to (n = 45).

The collected data from the 26 studies included are summarized in the Appendix A. Regarding the etiological factors associated with feline squamous cell carcinoma: (i) 19 articles identified and studied the following virus: feline PV, Human PV 38 (HPV38), feline immunodeficiency virus (FIV), feline leukemia virus (FeLV), feline syncytial virus, torque tenovirus (TTV), human herpesvirus type 4, and feline herpesvirus type 1 (FHV- 1); (ii) 4 articles studied the association between exposure to environmental tobacco smoke and the development of FOSCC; (iii) 4 articles investigated the association between FOSCC and diet, deworming methods, and other factors, such as the source of heating, water, and the environment in which the animals lived and exposure to sunlight, and (iv) 4 articles reported comorbidities with expression in the oral cavity that may be associated with the development of the disease.

The studies included in this critical review were carried out in the following countries: United States of America (n = 12), Italy (n = 7), United Kingdom (n = 2), New Zealand (n = 4), Slovenia (n = 1), and Japan (n = 1).

The selected studies evaluated, in their entirety, 669 cats with oral SCC, and the number of samples ranged between 1 and 113 individuals/samples. Breed could not be identified in 476 of the 669 cats. The remaining 158 were from pure breeds: Abyssinian (n = 1), Burmese (n = 1), Chartreux (n = 8), Himalayan (n = 6), Maine Coon (n = 5), Persian (n = 2), Siamese (n = 3), European Shorthair (n = 306), Korat (n = 1), and unspecified purebred cats (n = 2).

Information on the gender was only available for 437 cats, including 258 females and 233 males. Gender was not recorded/referred to for the remaining cats. Patient ages ranged between 2 and 20 years.

In seventeen studies that researched about viruses, four used only PCR (polymerase chain reaction), eight used PCR and immunohistochemistry (IHC), two used PCR and in situ hybridization (ISH), and one used PCR and ViroCap (next generation sequencing method). In two studies, the research method was not mentioned. Concerning the research of viral agents, 13 publications focused on the research of papillomavirus (PV).

In seven studies, PV detection was performed by PCR and IHC, in four by PCR, in one by IHC, PCR, and ISH, in one by PCR and ViroCap, and another by PCR and ISH. These 14 studies evaluated a total of 292 cases of FOSCC, with PV being detected in 43 cases (14.7%). The incidence of PV ranged between 0 and 100%. The most identified genotype was *Felis catus* papillomavirus-2 (FcaPV-2), and *Felis catus* papillomavirus-1 (FcaPV-1), -3 (FacPV-3), and -4 (FcaPV-4) genotypes have also been reported, as well as human papillomavirus type 38.

The association between exposure to environmental tobacco smoke and the development of FOSCC was described in three studies. From a total 185 cats with FOSCC, this exposure was verified in 61 animals (32.9%). In two of these four studies [12,13], p53 expression was analyzed by IHC, which was observed to be increased in 32 samples (68.1%) out of a total of 47.

In relation to the two studies in which the association between oral SCC and type of diet was evaluated, 159 cats were included [12,14]. About 58.5% of these animals ate canned food, 56.6% canned tuna, and only 22% ate dry diet. In these same two studies, the association between FOSCC and the administration of deworming products was also evaluated, and it was observed that in 26 of the 159 (16.35%) cats used deworming collars.

Regarding the relationship of oral SCC with other comorbidities, among the 485 cats with this disease referred to in the studies included in this CR, thirty-one cats (6.4%) had dental pathology, but only one was referred to with diagnosis of feline chronic gingivostomatitis; one cat had an infection by *Trichinella* spp. and another was reported with an osteosarcoma.

The methodological quality scores based on Downs and Black checklist ranged from 3 to 7, as presented in Appendix A.

## 4. Discussion

According to the authors’ knowledge, this is the first critical literature review that ascertains the etiology of the feline oral squamous cell carcinoma. There is, to date, evidence of some known etiological factors of the human head and neck squamous cell carcinoma (HNSCC), such as tobacco and alcohol consumption habits and Human papillomavirus (HPV) infection [10].

Nonetheless, and though these tumors share clinical, histological, and biological factors [8,9], there is a lack of published information available about the etiology of the FOSCC. Thus, the present publication has as a primary focus of the identification of possible described etiological agents for this neoplasia and to collate and summarize all the available information about this research question. Due to the reduced number of publications containing statistical data, a meta-analysis was not possible at this point. Concerning the possible viral etiology of the disease, PV was the most commonly virus scanned and identified in cats with oral SSC and on FOSCC samples. In humans, an association between HPV16/18 and HNSCC is described [15,16]. Similarly, some authors report a possible association between a chronic infection with canine papillomavirus and the malignant transformation of lesions in SCC, in particular, the transformation of oral papilloma in oral SCC [17,18,19,20].

Perhaps for that reason, the study of the role of PV in oral feline SCC has sparked the interest of many researchers, justifying the rise in the number of published works focusing on this topic since 2015. PV is a double-strand DNA virus, with no envelope, with tropism for stratified squamous epithelium of the skin and mucosa of humans and animals [21]. Recently, several types of feline PV were described and identified on oral tissues, in some cases related to several gingivitis [22], but with no evidence that is related to oral SCC in these animals [23,24], but it could be involved as an initiation step, and after this, with some environmental co-factors to promote these lesions [25].

The infection begins when the virus enters the cells of the basal layer and makes use of the cellular machinery to replicate, producing a small number of viral copies of round shape. In this way, the biological cycle of the virus is inherently related to the differentiation of epithelial cells [21].

Its carcinogenic role is attributed to its oncoproteins E6 and E7 that, respectively, degrade two tumor-suppressor proteins, namely, the guardian of the genome, also known as tumor protein p53, and the retinoblastoma protein (pRb), and thus promote aberrant cell proliferation and the initial steps of carcinogenesis [21]. In what concerns the feline PV in vitro, studies demonstrate the oncogenic properties of the virus through the p53 and pRb routes, particularly FcaPV-2 [26,27]. Additionally, the degradation of pRb results in an accumulation of protein 16 (p16), a cellular protein whose function is impeding the phosphorylation of pRb [21,28].

In that sense, the evaluation of the increase in cellular expression of p16 by immunohistochemistry (IHC) in samples of human HNSCC and of feline cutaneous SC has been used as a marker of the possible presence of feline PV [15,21]. Many studies were identified that were directed towards the role of PV in the carcinogenesis of FOSCC through the research of viral DNA and the analysis of the expression of oncogenes of PV by polymerase chain reaction [17,23,24,26,27,29,30,31,32,33]. Additionally, some authors also analyzed the expression of p16 by IHC attempting to diagnose PV infection [17,20,21,22,23,24,25,26,27,28,29,30,31,32,33,34,35].

Through PCR, it was possible to observe variability in the presence of PV DNA (0 to 100%) in the feline oral CE samples [17,23,24,29,30,32,33,34,35,36]. Altamura and collaborators (2020) [35], based on the findings, attributed PV an important role in the beginning of the development of SCC, because after the detection of FcaPV-2 in 31.3% of the 31 samples, they detected an expression of E6 and E7 oncogenes in 70% of the infected samples and an association between this with a high viral burden through PCR. Additionally, Altamura and colleagues (2022) [37] show the high prevalence of FcaPV-1/-2/-3/-4/-5/-6 in samples of FOSCC, confirming that type-2 is the most prevalent. The work of Skor (2015), presented at the Veterinary Cancer Society, also showed a high incidence of PV (12/12).

Despite this, and also using the PCR, the majority of the selected studies showed a low prevalence of PV in FOSCC (specifically in the search of PV in FOSSC, including only one sample of this kind of tumor, which contributes to the variability in incidence). Regarding the methodology used, the lack of detection of the virus might be related to the methodology used in the preservation of tissues, mostly formalin-fixed and embedded in paraffin (FFEP) tissues. While formalin allows for the preservation of DNA for many years, formaldehyde results in crosslinks between DNA and proteins, thus making the amplification of viral DNA harder [29,38] and sometimes degraded.

In humans, it is described that only 62.7% (xylene-treated) to 73.3% (heat-treated) of cases maintain an HPV positive status by PCR after the FFEP procedure, demonstrating that aggressive heat treatment results in higher DNA yields and increased sensitivity for HPV testing [39]. The discrepancy in the results obtained by Altamura et al. (2020) [35] in feline samples was attributed to the type of primer (consensus or specific) used in the PCR. The consensus primers amplify the region of the L1 gene of diverse genotypes of PV, some still unknown, while the specific primers have the capacity to detect specific genotypes [37]. However, the failure in detection of the virus does not seem to be related only to this factor, as some studies have applied both types of primers without detecting the virus in either sample.

Additionally, the studies of Skor (2015) and Altamura and colleagues (2020) used consensus and specific primers, respectively, and successfully detected the virus. It is, nonetheless, possible that the samples were infected by genotypes not detectable by the primers employed in each study, and many employed only the JMPF/R primer, which is specific to FcaPV-2 [30,40,41]. Another factor to highlight is the possibility of the occurrence of the “hit and run” mechanism by which the virus can initially be present in the lesion and induce cellular transformation, and later disappear [42]. In those cases, viral DNA could be undetectable in the lesion it originated [29,40]. It is important to highlight, thus, that the lack of detection of the viral DNA does not exclude the possibility of the virus being involved in the development of the tumor [29]. As such, and considering the facts previously described, the number of PV positive cases might be underestimated.

In some cases, DNA sequencing was employed to determine the genotype of the PV present, resulting in an identification of FcaPV-2 in most positive samples. This result was expected, as a huge majority of the studies made use of the specific JMPF/R primer [30,40].

On the other hand, the detection of PV is not enough to establish a causal association between feline PV and FOSCC. An example of this is the detection of PV DNA in the oral cavity of a cat presenting only gingivitis [22]. In this way, to demonstrate that PV is responsible for the development of FOSCC, besides detecting viral DNA in the tumor samples it is necessary to detect the expression of oncogenes E6 and E7, as well as to demonstrate the interaction of the viral oncoproteins with tumor suppressor genes such as TP53 and RB1 [34,43]. Out of the six articles evaluating the expression of p16 by IHC in the FFEP samples of oral SCC, five [17,23,30,31,35] resorted to the same antibody (mouse monoclonal antibody anti-p16INK4a, clone G175-405, BD Biosciences, San Jose, CA, USA), with the exception of the Yamashita-Kawanishi et al. (2018) study that used the mouse monoclonal antibody anti-p16INK4a, from Becton Dickinson Company, Franklin Lakes, NJ, USA [32]. The use of a different antibody, and different laboratory conditions, are in themselves reasons why the results obtained could be different. The IHC identifies the protein produced, not the way in which it is produced, so conclusions cannot be drawn about who produced it and in what way. Neither of the studies found a statistically significant correlation between the increasing in this protein and the infection by PV. These results suggest that the overexpression of p16 was not caused by the virus [23]; nonetheless, one cannot exclude the possibility of a failure in detection of the virus in samples with overexpression of p16 [30]. p16 is considered an effective biomarker of cell senescence and molecular aging, given that it increases with age and encodes for a protein that blocks cyclin-dependent kinase, which promotes cellular senescence and permanent cell-cycle arrest [44].

The feline immunodeficiency virus (FIV) and the feline leukemia virus (FeLV) belong to the Retroviridae family and are responsible for the immunosuppression of infected cats, leaving them more susceptible to secondary and chronical infections, as well as the development of neoplasia [38]. FeLV is recognized as a cause of feline lymphoma and leukemia, although its incidence in the general cat population is still unknown [45]. An association between lymphoma, squamous cell carcinoma, fibrosarcoma, and mastocytoma with infections by FIV is reported [46], however, in the present review, only two studies related to FIV in FO SSC samples, through PCR and ViroCap, respectively. In the remaining studies, the viral diagnostic methods were not described, mentioning only the sanitary status of the animals.

The results obtained in this revision seem to demonstrate a low incidence of this virus in cats with oral SCC, however, this result could be attributed not only to the lack of information provided about the sanitary status of the animals, but also to the low number of studies that looked for this virus. Concerning FeLV, the investigation of this virus in oral SCC is sparse, as only two of the studies found searched for this virus in samples of oral SCC through ViroCap [34] or through another unmentioned diagnostic technique [47]. The means of diagnosis are important to consider in the interpretation of results, as the search for the virus through serological tests during the regressive phase of infection might originate false-negative results [38].

The Epstein–Barr oncogenic virus (EBV) has tropism for B cells, but can also infect epithelial cells. Infection by EBV is considered a risk factor for the development of HNSCC in humans. Infection by EBV in cats and dogs was described by Milman et al. (2011) [48], but in this revision, this virus was assessed in only one study through ViroCap and PCR, resulting in one positive cat (out of twenty), coinfected with FcaPV-3 [34]. Coinfection with both these viruses is also described in humans with HNSCC and is connected to advanced stages of the disease [49]. Cohabitation and close contact of cats with humans enables their exposure to the same environmental factors, be it through ingestion during grooming, or through inhalation [14,50].

In humans, tobacco smoking and the exposure to environmental tobacco smoke (ETS) are the main risk factors for the development of oral cancer [51]. Tobacco contains around 60 substances capable of inducing epigenetic alterations in cells of the oral epithelium, inhibiting functions of the immune system, causing oxidative stress in the tissues, and consequently contributing to the development of oral SCC [51]. The research teams coordinated by Bertone (2003) [14], Snyder (2004) [12], and Renzi (2019) [13] investigated the hypothesis of the relation between exposure to ETS and the development of FOSCC.

Resorting to questionnaires directed at owners of cats with an FOSCC diagnosis, information was collected about exposure to tobacco smoke, for example, the number of years of exposure, number of smokers in the household and of cigarettes consumed daily. Contrary to what is found in humans, neither of these studies found a statistically significant correlation between ETS and the development of the tumor, even though in the study by Bertone and collaborators (2003) [14] the risk of developing oral SCC was two times higher in cats exposed to ETS when compared to those never exposed.

There was not an evident dose–response relation either: there was no correlation found between the number of cigarettes consumed by the owners and the development of FOSCC.

Many studies focused on humans demonstrate that the consumption of tobacco results in altered function of the p53 protein, codified by the tumor suppressor gene TP53, which takes on a fundamental role in the control of cell cycles, avoiding the replication of cells presenting DNA anomalies. The mutation in the TP53 gene is the most common alteration found in the carcinogenesis of HNSCC in humans. In human medicine, many studies relate the expression of this protein (assessed by IHC) and exposure to risk factors such as tobacco consumption [52,53]. Out of the studies investigating the exposure to ETS in cats with oral SCC, only two analyzed the expression of p53 in oral tissue and its association with exposure to ETS [12,13].

Results obtained by Snyder and collaborators (2004) did not show statistical significance (*p* = 0.19), but a tendency for the rise in the expression of p53 (4.5 times) in cats exposed to ETS was observed [13]. These researchers, however, do not describe any relation between the mutation of the gene TP53, identified in 70% of the cases, and exposure to ETS [13]. It is important to highlight that the rise in the expression of p53 could also be related to other factors besides exposure to ETS, such as infection by oncogenic or other viruses.

Two studies investigated the relation between ectoparasite control methods and FOSCC, with no statistically significant results [12,14]. Bertone and colleagues (2003) [14] suggested that there is a 5.3-fold risk of developing oral SCC in cats that use a deworming collar.

Bertone (2003) and Snyder (2004) [12,14] also investigated the potential etiology of FOSCC being associated with the diet. Although the results were not statistically significant, the first suggest that cats that eat a mostly wet diet are 3.6 times more likely to develop this tumor. It should be noted that a significant percentage of cats with oral SCC eat canned food (50.9%), mainly tuna fish (45.9%), so there seems to be some related risk. Snyder and colleagues (2004) did not find an association between the factors previously described and the expression of p53. The existing evidence for an environmental etiology of FOSCC is, therefore, very limited. The reduced size of the samples used in the studies by the teams of Snyder (2004) [12] (n = 23), Renzi (2019) [13] (n = 24), and Bertone (2003) [14] (n = 36) may contribute to this, limiting the statistical analysis. In addition, the incomplete participation of the cat owners included in these studies may lead to a selection bias, since the group of owners who agreed to answer the questionnaires may not represent the real number of smoking owners with cats that developed FOSCC.

In several studies on FOSCC, the presence of other comorbidities in the oral cavity is described [54,55,56] among them, tooth exfoliation, periodontal disease (PD), feline chronic gingivostomatitis, tooth resorption lesions, and infection by *Trichinella* spp.

In this review, the number of cats with oral SCC and periodontal disease seems to be underestimated, as periodontal disease is a very common condition in the feline oral cavity, especially in older animals [57]. This may be due to the lack of information available on oral and dental examination in the selected studies. Feline chronic gingivostomatitis (FCGS) is a chronic inflammatory disease that is very common in the oral cavity of cats [58] and was only described by Olmsted et al. (2016) [55], who report a cat with both comorbidities. Therefore, to date, there are no published studies investigating and describing an association between periodontal disease, chronic inflammation, and the development of FOSSC.

On the other hand, this association is the subject of intense study in humans. Gopinath et al. (2020) [59] suggest that PD is a risk factor for the development of HNSCC, due to the chronicity of the inflammatory and infectious process. Chronic inflammation can lead to genetic and epigenetic changes and cause the suppression of tumor suppressor genes, such as, for example, the TP53 gene, common in HNSCC in humans. In FOSCC samples, mutations in this gene and the dysregulation of p53 expression are described [9,12,23,60,61].

During the inflammatory process, mediators such as cytokines, prostaglandins, and matrix metalloproteinases (MMP) are released and pro-inflammatory transcription factors are activated, such as NF-κB and the transducer and transcription activator signal 3 (STAT3), which play an important role in suppressing apoptosis, proliferation, angiogenesis, invasion, and tumor metastasis. Some studies, in vivo and in vitro, have demonstrated the deregulation of COXs [6], STAT3 [62], EGFR [63], VEGF [64], and CD147 [65] in FOSCC samples.

Additionally, some authors believe that periodontal–pathogenic bacteria are involved in three phases of the development of HNSCC: in the epithelial–mesenchymal transition, in neoplastic proliferation, and in tumor invasion [66]. In this review, a case of nematode infection by *Trichinella* spp., in a sample of FOSCC, was included, not attributing an etiological role to the parasite [54]. However, in humans, an association between this parasitosis and oral carcinoma has been described, although its role in the development of the tumor is still unclear [67], that is, it has not yet been possible to distinguish whether the infection by *Trichinella* spp. is a carcinogenic factor or an occasional finding in the lesion. During the life cycle of this parasite, there is a release of larvae that migrate to the lymphatic system and blood vessels and then penetrate the skeletal muscle, where they grow in the muscle fibers. Some authors suggest that local irritation caused by chronic muscle inflammation secondary to infection may contribute to the development of malignant oral carcinoma [67]. Quigley and collaborators (1972) [68] describe the existence of oral SCC in two cats and suggest that this neoplasia originated from tissue with dental embryonic origin, namely, from the dental lamina and the epithelium of the enamel organ. In human medicine, some cases of oral SCC originating from dentigerous cysts are also described [69].

In this review, information on the gender, cat’s breed, and life stage was not available for all cats and no relationship was found between these parameters and the evaluated etiologic factors.

It is also important to mention some of the limitations of this review. One of them is the small number of published studies related to the etiology of FOSCC. In addition, this review mostly included studies with small sample sizes, the majority of which are studies of case series. The restriction of the articles choice to English, Portuguese, Spanish, and French, and the fact that the research was carried out only in one database (PubMed), may have contributed to the loss of information relevant to this study. In addition, it is not possible for authors to assure that some cats were not counted twice, since studies included do not provide this information.

It was also not possible to access the full text of some works potentially relevant to the study. In our opinion, the present study has high scientific interest, as it gathers information from several original studies, identifies the gaps in the knowledge of the etiology of FOSCC and, at the same time, provides information potentially relevant to future clinical investigations. In addition, we believe that we have established useful comparisons between FOSCC and human HNSCC, the sixth most common type of cancer in the world, about which much remains to be investigated in terms of etiology, pathogenesis, treatment, and prognosis [70]. These two neoplasia share similar clinical and molecular markers and biological behavior. For this reason, cats with FOSCC may serve as a spontaneous animal model for the study of HNSCC [8].

Increasingly, domestic animals and humans share the same space, are exposed to the same environmental factors, and chemical and infectious agents, and in this way, cats can play a very important role as sentinels of the disease in the human population [50,71,72,73,74].

## 5. Conclusions

Viruses can play a crucial role in oral carcinogenesis, due to its prooncogenic effect or by triggered immunosuppression which allows neoplastic proliferation. There is not enough evidence for the oncogenic role of PV in FOSCC or the hypothesis for using p16 expression as a marker for the presence of PV in this tumor.

Chronic exposure to environmental factors may be at the origin of FOSCC, also conjugated with virus etiology. The bibliography has a hypothesis that cats feed with canned food and those which use deworming collars have a higher risk to develop FOSCC.

Periodontal diseases and chronic inflammation in the feline oral cavity are frequent, and in analogy with what has been described for humans, it is important to understand their relationship with FOSCC.

The results we obtained showed that the available evidence on the etiology of FOSCC is still limited, however, there is a need to expand studies in this area, given the increasing interest in this topic.

## Figures and Tables

**Figure 1 vetsci-09-00558-f001:**
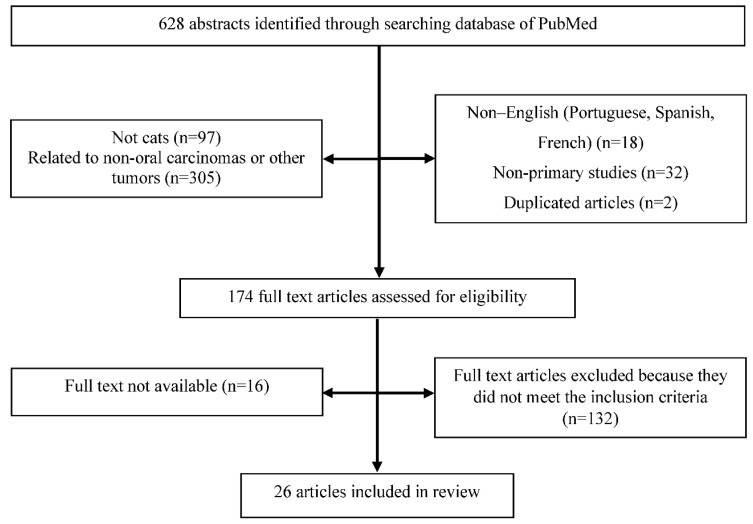
PRISMA flow chart for study selection.

## Data Availability

Not applicable.

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
