# Peer review of "Feline Oral Squamous Cell Carcinoma: A Critical Review of Etiologic Factors"

_vetsci, 2022, doi:10.3390/vetsci9100558_

Round 1

Reviewer 1 Report

Sequerira et al provides a systematic review of etiologic factors for the development of feline oral squamous cell carcinoma (FOSCC). FOSCC is an important veterinary disease and a better understanding of pathogenesis and treatment is worthy of study and I believe this topic will be of interest to readers. The authors describe methods for selecting research papers, summarize their findings, and offer discussion. The overall strategy appears to involve combining individual data from multiple studies. The paper would be strengthened by a better justification and explanation for that approach, with reference to consultation with a statistician, and presentation of descriptive statistics for the various risk factors in a dedicated table or graph. Currently, each separate paper is described, but a table with the overall results would be helpful, assuming the strategy is appropriately guided by someone familiar with this form of analysis (this reviewer is knowledgeable of the subject, but not the method and scientific rigor of combining individual data in this manner). Once the methods and results are more clearly presented, the discussion, introduction and abstract will require revision. The authors should revisit the literature for recent publications that should be included in the discussion and/or the introduction.

Comments are organized by section.

1.      Introduction:

-        Expand to include more background. It is currently shorter than the abstract. Introduce broad categories of possible etiologies in feline cancer, and in human OSCC, to set the stage. Also, the need for this systematic review should be more clearly stated. What will this review provide to the field? The overall strategy should be introduced and clearly justified.

-        There is a reference / citation issue in the introduction that carries forward throughout the manuscript. Primary literature should be referenced whenever possible rather than text books or the introductions of other primary research papers that refer to someone else’s work. For example, lines 59-61 contain at least 2 references that do not appear to have studied treatment and prognosis, instead, they likely made a similar statement in their introduction.

-        Reference for line 56 is a primary research paper looking at a novel treatment in a mouse model of FOSCC. It seems like there should be a better reference for supporting a general statement on the growth rate and destructive nature of this tumour in cats.

2.      Materials and Methods:

-        Citations for PRISMA and Cochrane Collaboration? This opening paragraph should briefly explain how the data was analyzed (individual data from various studies were combined) and the reasoning for it. Is this considered Integrative Data Analysis?

-        2.1: Were there relevant articles that were missed because the journal was not indexed by pubmed? Why was ‘Gingiva’ not included as a search term? It is a common location for OSCC. If ‘Lingual’ and ‘Labial’ were specified search terms, why not other anatomic locations? Could this have affected the outcome?

-        2.2: Why were labial samples included? It seems plausible that etiology may be different at this site (solar radiation) compared to intraoral sites. The significance should be explained.

-        What does it mean that papers were excluded if they were not available online? Were other strategies explored for securing unavailable papers such as an interlibrary loan?

-        2.4: Is the quality of the study the same as the level of evidence? What were the 9 criteria used to assess quality? How were quality and level of evidence used in the interpretation of the results?

3.      Results:

-        The flow chart is a nice addition to the paper.

-        Line 156 states papers were excluded because an etiologic factor wasn’t studied. What about papers that investigated breed, castration status, inflammation or tumour location? While not specific etiologic agents, these can often hold clues that hint at possible risk factors that warrant further study.

-        The abstract states that papers published between 2000 and 2020 were considered, but this criteria is not mentioned in the methods, and 3 papers are included in table 4 that are earlier than 2000.

-        Tables would be more helpful if significant findings and trends were highlighted. Some of the columns are cluttered and hard to read.

-        Table 4: Some have risk factors listed in the last column, but there is no %, RR or p value.  

-        Some of the information captured in the tables isn’t described or discussed in the text. What is the significance of article quality? Sex of cat? Age? Breed?

-        How many of these studies included control cats? Or cats without FOSCC? The tables have a column for the number of cats with FOSCC, but if there arent cats without FOSCC, or some measure of the general feline population, how can a risk factor be associated with a disease? For example, Bertone et al used 112 feline patients with renal failure as a control group. That information is just as important as the number of patients with FOSCC, there should be a column in in the tables to address controls.

-        Most of the results are arranged as summary statements of the proportion of cats with FOSCC that had a risk factor / etiologic agent. There is no mention of what the frequency of these risk factors / etiologic agents were in the control groups, or some other measure of the general feline population. It isn’t very helpful to state how many FOSCC cats had ETS exposure, wore flea collars, or ate canned food, if we don’t know how common these things are in the control population.

-        The discussion explains why a meta-analysis was not performed, but in the results, data from papers is combined together to give overall statistics (percentage of PV positive cases for example). Was a statistician involved?

-        Were any of the cats counted twice? For example, it appears many FOSCC cases in the Bertone 2003 survey might be the same as those described in the Snyder 2004 survey (same institution, overlapping time period, published only a year apart).  

4.      Discussion

-        Line 243: Good to point out limitations preventing meta-analysis. Should include appropriateness of combining individual cats across various studies.

-        Line 259-265: Please clarify. One part of the sentence says there is no evidence that PVs are related to  OSCC, but the other part says that PVs might interact with environmental factors to promote OSCC.

-        Do E6 and E7 directly degrade p53 and pRb? Should be clarified.

-        Paragraph starting at Line 297: Rephrase to acknowledge the high degree of variability (0-100% suggests methods are not very reliable). Altamura’s 2020 paper showing greater than 30% of samples were positive seems quite high. They have a more recent paper (that should be included and referenced in the discussion) which shows FcaPV-2 is positive in fewer cases. That should be discussed even though the recent paper is too new to be captured in the analysis.

-        Did the groups showing PV infection in FOSCC also evaluate PV infection in healthy oral tissues? Or oral biopsies without FOSCC? Was it associated with location? (human OSCC associated with PV infection is more often pharyngeal, while cats are known to get cutaneous SCC (Bowenoid carcinoma in situ) due to PV). Are the positive cases of FOSCC of the lip? Or were they tongue or gingiva?

-        Line 322: Numbers need to be corrected

-        Paragraph starting at line 322 offers interesting discussion, but needs to be clarified. How would consensus primers lead to artificially low measures of PV infection? Generally speaking, lack of specificity makes false positives more likely.

-        Discussion starting on line 340 regarding hit and run hypothesis is helpful. How does that relate to PCR for E6/E7 vs L1/L2?

-        Discussion should include how these studies validated their primers and results. Were appropriate controls used? Did they sequence the amplicons to confirm they were amplifying what they thought they were?

-        375-378: Good mention of cases that are high P16 but no evidence of PV infection. Is there any other explanation besides failure to detect PV? (Are there other reasons that p16 could be increased?) Were P16 IHC methods validated?

-        379 – 386: A reasonable paragraph, though it should be acknowledged that the focus is on studies using clinical samples. If the conclusion is that more mechanistic studies of the role of PV in FOSCC are needed, then there should be some comment on in vitro studies using cell lines. (it was already stated that in vitro studies were excluded from the systematic review).

-        All discussion of viruses in FOSCC should include discussion of these viruses in the general cat population, known or estimated, otherwise there is no way to know if the viruses are a risk factor.

-        Paragraphs on ETS, flea collars, diet need to be clarified a little. They say there is no statistically significant relationship, then state that risk was increased. This should be expanded a little to discuss ‘trends’ vs. ‘statistically significant association’. Otherwise they sound like contradictory statements.

-        479-481: “It should be noted that a significant percentage of cats with oral SCC eat canned food (50.9%), mainly tuna fish (45.9%), so there seems to be some related risk.” This must be interpreted in the context of the general population. Many cats eat canned food, many cats eat tuna fish. High incidence of this diet in the FOSCC cases does not equate to increased risk.

-        484-491: Good mention of small sample sizes. How could selection bias impact all of the studies? What type of cats are captured by these studies?

-        498: Good mention of likely under representation of stomatitis

-        520-529: Although Nasry 2018 is cited in the text regarding CD147, it does not appear in the reference list.  Inflammation in human and animal OSCC (including FOSCC) has been recently reviewed, consider including Nasry 2021 in this section.

-        547: Chronic muscle inflammation or chronic mucosal inflammation?

-        561: The authors state some papers may have been missed by only using pubmed. Does using another database reveal more papers? Has that been checked?

-        576: Sounds like the authors are making a statement about their own work, then cite Wypij et al. This could be rephrased.

Abstract: suggest a rewrite after results of manuscript are clarified, it should be more succinct and not overstate significance of findings.

Author Response

Sequerira et al provides a systematic review of etiologic factors for the development of feline oral squamous cell carcinoma (FOSCC). FOSCC is an important veterinary disease and a better understanding of pathogenesis and treatment is worthy of study and I believe this topic will be of interest to readers. The authors describe methods for selecting research papers, summarize their findings, and offer discussion. The overall strategy appears to involve combining individual data from multiple studies. The paper would be strengthened by a better justification and explanation for that approach, with reference to consultation with a statistician, and presentation of descriptive statistics for the various risk factors in a dedicated table or graph. Currently, each separate paper is described, but a table with the overall results would be helpful, assuming the strategy is appropriately guided by someone familiar with this form of analysis (this reviewer is knowledgeable of the subject, but not the method and scientific rigor of combining individual data in this manner). Once the methods and results are more clearly presented, the discussion, introduction and abstract will require revision. The authors should revisit the literature for recent publications that should be included in the discussion and/or the introduction.

R: We are very grateful for your comments and thoughtful suggestions that surely lead to improvement of the quality of the manuscript.

Comments are organized by section.

  1. Introduction:

- Expand to include more background. It is currently shorter than the abstract. Introduce broad categories of possible etiologies in feline cancer, and in human OSCC, to set the stage. Also, the need for this systematic review should be more clearly stated. What will this review provide to the field? The overall strategy should be introduced and clearly justified.

R: The Introduction section was improved according to the reviewer's suggestions.

- There is a reference / citation issue in the introduction that carries forward throughout the manuscript. Primary literature should be referenced whenever possible rather than text books or the introductions of other primary research papers that refer to someone else’s work. For example, lines 59-61 contain at least 2 references that do not appear to have studied treatment and prognosis, instead, they likely made a similar statement in their introduction.

R: The reference Bilgic (2015) was removed from the manuscript because this paper is a review. A new reference (Hayes et al., 2007) was added.

- Reference for line 56 is a primary research paper looking at a novel treatment in a mouse model of FOSCC. It seems like there should be a better reference for supporting a general statement on the growth rate and destructive nature of this tumour in cats.

R: The references Martin et al. (2011 and 2015) were removed from the manuscript, because these studies were focused on other species (rodent). New references (Mikiewicz et al., 2019; Vail et al., 2019) were added.

  1. Materials and Methods:

- Citations for PRISMA and Cochrane Collaboration? This opening paragraph should briefly explain how the data was analyzed (individual data from various studies were combined) and the reasoning for it. Is this considered Integrative Data Analysis?

R: After better analyzing and reviewing the experimental design of the work, we considered that this study does not respect some norms inherent to a qualitative systematic review. Thus, we suggest that this work can be defined as a critical review. We emphasize that it was not possible to carry out a statistical analysis, since only a small number of studies included inferential statistical data which could be subjected to quantitative systematic (meta-analysis) analysis. The manuscript was updated accordingly.

- 2.1: Were there relevant articles that were missed because the journal was not indexed by pubmed? Why was ‘Gingiva’ not included as a search term? It is a common location for OSCC. If ‘Lingual’ and ‘Labial’ were specified search terms, why not other anatomic locations? Could this have affected the outcome?

R: A new bibliographic search was performed, by using the same protocol and adding the MeSH entry "Gingiva" and the same references were found. The literature search was updated accordingly, by adding a new MeSH entry. The initial search was performed on PubMed which contains only indexed articles. The selected articles are also part of the Web of Science database which the authors, however, did not use in the initial research.

- 2.2: Why were labial samples included? It seems plausible that etiology may be different at this site (solar radiation) compared to intraoral sites. The significance should be explained.

R: Labial location was included in the present study, because it is considered that the anatomical and histological limits of the oral cavity is the mucocutaneous junction. During the analysis of the selected articles, all tumors from cutaneous origin were discarded. The solar radiation as an etiological factor of SCC was described by Dorn and collaborators (1971). These authors considered the neoplasms found in the “lip” as being cutaneous and not from the oral cavity, although different oral locations were reported in this publication. Due to the fact that solar radiation was not related to oral neoplasms, the authors decided to do not select the publication for analysis. The authors thanks this important comment.

- What does it mean that papers were excluded if they were not available online? Were other strategies explored for securing unavailable papers such as an interlibrary loan?

R: We avoid using only abstract information. All the works were read in their entirety, therefore, when we do not have the complete work (which was obtained by open access papers, or the signature of our institution), we discarded this work, as we did not have complete information on it. The authors have the opinion that any document not available for free access is not useful for society, namely for clinicians who do not have access to bibliographic databases with academic or institutional access.

- 2.4: Is the quality of the study the same as the level of evidence? What were the 9 criteria used to assess quality? How were quality and level of evidence used in the interpretation of the results?

R: The authors stated: “In order to assess the studies, only nine items were considered. The remaining items were excluded as there were no experimental studies with statistical power“. Due to the fact that this article evaluate responses to treatments, it was not possible to fully apply the Downs and Black scale. For this reason, it was necessary to adapt it using the analysis of nine criteria. The Results section was updated accordingly.

  1. Results:

- The flowchart is a nice addition to the paper.

R: The authors thank the reviewer’s comment.

- Line 156 states papers were excluded because an etiologic factor wasn’t studied. What about papers that investigated breed, castration status, inflammation or tumour location? While not specific etiologic agents, these can often hold clues that hint at possible risk factors that warrant further study.

R: The authors also extracted information about breed, gender and age of the animals, and inflammation or tumour location as well from the selected articles.

- The abstract states that papers published between 2000 and 2020 were considered, but this criteria is not mentioned in the methods, and 3 papers are included in table 4 that are earlier than 2000.

R: A new bibliographic search of publications later than 2020 was performed. Fifty-six new references was founded and the analysis using the same protocol. Figure 1 (PRISMA flow chart for study selection) was updated accordingly.

- Tables would be more helpful if significant findings and trends were highlighted. Some of the columns are cluttered and hard to read.

R: The tables were improved aiming to upgrade the reading.

- Table 4: Some have risk factors listed in the last column, but there is no %, RR or p value.

R: From some selected bibliographic references, it was not possible to extract statistical data, such as %, RR or p value.

- Some of the information captured in the tables isn’t described or discussed in the text. What is the significance of article quality? Sex of cat? Age? Breed?

R: The frequency of the cats’ breed and sex are described in the Results section. Information about the life stage was added. No relationship was found between these parameters and the evaluated etiologic factors.

- How many of these studies included control cats? Or cats without FOSCC? The tables have a column for the number of cats with FOSCC, but if there arent cats without FOSCC, or some measure of the general feline population, how can a risk factor be associated with a disease? For example, Bertone et al used 112 feline patients with renal failure as a control group. That information is just as important as the number of patients with FOSCC, there should be a column in in the tables to address controls.

R: The authors did not refer to data from control groups, because the majority of the selected articles, except the published by Bertone and collaborators (2003), were not case-controlled.

- Most of the results are arranged as summary statements of the proportion of cats with FOSCC that had a risk factor / etiologic agent. There is no mention of what the frequency of these risk factors / etiologic agents were in the control groups, or some other measure of the general feline population. It isn’t very helpful to state how many FOSCC cats had ETS exposure, wore flea collars, or ate canned food, if we don’t know how common these things are in the control population.

R: The authors are aware about the limitations of studies based on questionnaires. It is not possible to describe cats exposed to ETS or the general feline population. In the future, prospective case-control studies are mandatory to evaluate the effect of several etiological factors on the development and evolution of FOSCC. Unfortunately, as the authors stated, the literature on the subject is very heterogeneous and our study allows to enhance these limits and stresses the need of future large multi-institutional case control studies. 

- The discussion explains why a meta-analysis was not performed, but in the results, data from papers is combined together to give overall statistics (percentage of PV positive cases for example). Was a statistician involved?

R: One of the co-authors (José Leitão) has expertise in statistical analysis, in particular focusing on both qualitative and quantitative bibliographic reviews (https://orcid.org/0000-0003-1798-2496).

- Were any of the cats counted twice? For example, it appears many FOSCC cases in the Bertone 2003 survey might be the same as those described in the Snyder 2004 survey (same institution, overlapping time period, published only a year apart).

R: This is a very pertinent comment, thank you. Unfortunately, we cannot be sure of that as methodology sections of these publications do not provide this information, and the raw data is not available. Thus, it was not possible for the authors to assure that some cats were not counted twice. This comment was added to the Discussion.

  1. Discussion

- Line 243: Good to point out limitations preventing meta-analysis. Should include appropriateness of combining individual cats across various studies.

R: As stated in the manuscript, the authors considered the lack of inferential statistical analysis as a limitation for meta-analysis. We agree with the reviewer that the duplication of animals between studies could cause a bias in the analysis.

- Line 259-265: Please clarify. One part of the sentence says there is no evidence that PVs are related to OSCC, but the other part says that PVs might interact with environmental factors to promote OSCC.

R: As referred by Gil da Costa and colleagues (2016), cited in our manuscript, the environmental co-factors are thought to cooperate with papillomaviruses in facilitating viral persistence and promoting the development of viral lesions. These effects are mostly attributed to immunotoxin agents that hamper an effective immune reaction and viral clearance and to genetic damage caused by genotoxic agents. In the authors’ knowledge, although already identified a tropism for stratified squamous epithelium of the skin and mucosa, there is no evidence about the mechanism for induction of oral lesions, such as gingivitis in animals.

- Do E6 and E7 directly degrade p53 and pRb? Should be clarified.

R: Discussion was improved by adding that “tumour supressor proteins namely the guardian of the genome also known as tumor protein P53 (p53)”. The literature mention that “In the middle layer, as the epithelium is differentiated, the TFs that induce viral genome amplification are expressed, inducing that HPV early proteins increase their expression. Thus, the levels of HPV E6 and E7 oncoproteins increase, and the interactions of these oncoproteins with p53 and pRB, respectively, are induced. The latter induces the degradation of p53 and pRB, preventing cell cycle arrest and cell death via apoptosis but provoking cell cycle activation to progress from G1 to the S phase.”.

- Paragraph starting at Line 297: Rephrase to acknowledge the high degree of variability (0-100% suggests methods are not very reliable). Altamura’s 2020 paper showing greater than 30% of samples were positive seems quite high. They have a more recent paper (that should be included and referenced in the discussion) which shows FcaPV-2 is positive in fewer cases. That should be discussed even though the recent paper is too new to be captured in the analysis.

R: The manuscript was improved, by discussing the most recent results published by Altamura on the assessment of PV on tissue samples by PCR.

- Did the groups showing PV infection in FOSCC also evaluate PV infection in healthy oral tissues? Or oral biopsies without FOSCC? Was it associated with location? (human OSCC associated with PV infection is more often pharyngeal, while cats are known to get cutaneous SCC (Bowenoid carcinoma in situ) due to PV). Are the positive cases of FOSCC of the lip? Or were they tongue or gingiva?

R: Chu and collaborators (2019) tested the hypothesis that viruses would be enriched in FOSCC compared to normal oral mucosa. However, PV was not found in samples from normal or clinically healthy oral tissue.

- Line 322: Numbers need to be corrected

R: The authors thank the reviewer. Numbers were corrected (62.7% and 73.3%).

- Paragraph starting at line 322 offers interesting discussion, but needs to be clarified. How would consensus primers lead to artificially low measures of PV infection? Generally speaking, lack of specificity makes false positives more likely.

R: The relative frequencies of HPV detection by PCR were clarified by adding the method used for paraffin removal (by xylene or by tissue lysis at 56°C). The cited reference mention that the aggressive heat treatment demonstrated an advantage over traditional xylene purification protocols, resulting in higher DNA yields and increased sensitivity for HPV testing. The authors agree that low specificity is related to false positives.

- Discussion starting on line 340 regarding hit and run hypothesis is helpful. How does that relate to PCR for E6/E7 vs L1/L2?

R: It is possible that PV causes FOSCC via a hit and run mechanism, where E6 and E7 contribute to tumor initiation, and the virus' genome is lost after accumulating mutations.

- Discussion should include how these studies validated their primers and results. Were appropriate controls used? Did they sequence the amplicons to confirm they were amplifying what they thought they were?

R: As referred by Altamura and colleagues (2018), the primers were selected as the most validated in the literature and employed to amplify a fragment of FcaPV-1 E1 (175 bp) and FcaPV-2 L1 genes (177 bp), respectively, and the melting curve analysis was performed and the melting peak compared with that obtained in the positive control to confirm the specificity of the amplicon.

- 375-378: Good mention of cases that are high P16 but no evidence of PV infection. Is there any other explanation besides failure to detect PV? (Are there other reasons that p16 could be increased?) Were P16 IHC methods validated?

R: The sentence was improved by adding “The p16 is considered an effective biomarker of aging. It increases with age and encodes for a protein that blocks cyclin-dependent kinase which promotes cellular senescence and permanent cell-cycle arrest.” Moreover, p16 expression is a quantitative measurement of senescence, a central mechanism by which environmental, genetic, and lifestyle damage affects the aging of an individual and leads to functional decline. A new article (Muss et al., 2020) was added to the manuscript and the reference list updated accordingly. Conversely, activation of p16 through reactive oxygen species, DNA damage, or senescence leads to the buildup of p16 in tissues and is implicated in the aging of cells.

- 379 – 386: A reasonable paragraph, though it should be acknowledged that the focus is on studies using clinical samples. If the conclusion is that more mechanistic studies of the role of PV in FOSCC are needed, then there should be some comment on in vitro studies using cell lines. (it was already stated that in vitro studies were excluded from the systematic review).

R: The authors thank the reviewer’s comment. Indeed, the focus of the present critical review was the etiological factors of the FOSSC, and not its physiopathology or the mechanism of action of etiological factors on the disease. Thus, we considered to erase this paragraph from the manuscript.

- All discussion of viruses in FOSCC should include discussion of these viruses in the general cat population, known or estimated, otherwise there is no way to know if the viruses are a risk factor.

R: The Discussion was improved by adding “FeLV is recognized as a cause of feline lymphoma and leukaemia; however its incidence in the general cat population is still unknown (Beatty et al., 2021)“. A recent publication in the journal Veterinary Sciences (Rolph & Cavanaugh, 2022; https://doi.org/10.3390/vetsci9090467) explores the association of infectious diseases in domestic cats and neoplasia, highlighting the need for further investigations in this area.

- Paragraphs on ETS, flea collars, diet need to be clarified a little. They say there is no statistically significant relationship, then state that risk was increased. This should be expanded a little to discuss ‘trends’ vs. ‘statistically significant association’. Otherwise they sound like contradictory statements.

479-481: “It should be noted that a significant percentage of cats with oral SCC eat canned food (50.9%), mainly tuna fish (45.9%), so there seems to be some related risk.” This must be interpreted in the context of the general population. Many cats eat canned food, many cats eat tuna fish. High incidence of this diet in the FOSCC cases does not equate to increased risk.

R: The authors thank the reviewer’s comments. As mentioned by the author, “exposure to household environmental tobacco smoke was associated with a nonsignificant 2-fold increase in risk (P 0.11).” Indeed, the described findings have no statistical significance, likely due to small sample size which reduces the statistical power. However, the authors consider the described finding important and warrant further investigation.

484-491: Good mention of small sample sizes. How could selection bias impact all of the studies? What type of cats are captured by these studies?

R: Studies of Snyder (2004), Renzi (2019) and Bertone (2003) are retrospective studies that included cats with FOSCC, whose owners were interviewed by e-mail or phone to collect information, including pet’s care and home environment. Incomplete participation by all eligible subjects can result in selection bias if participation is influenced by both exposure and disease status.

498: Good mention of likely under representation of stomatitis

R: Thanks for your comment. The affirmation is based on the evidence that oral inflammatory diseases are quite frequent in the feline population.

520-529: Although Nasry 2018 is cited in the text regarding CD147, it does not appear in the reference list. Inflammation in human and animal OSCC (including FOSCC) has been recently reviewed, consider including Nasry 2021 in this section.

R: The manuscript was improved, by replacing the most recent evidence published by Narsy (2021). This new article was added and the reference list updated.

547: Chronic muscle inflammation or chronic mucosal inflammation?

R: The authors intended to refer to the chronic muscle inflammation caused by T. spiralis which is associated with the development of malignant oral carcinoma both in feline and in human patients. In fact, other parasites are also related to the development of neoplasms in other tissues, such as Spirocerca lupi and esophageal sarcoma in dogs and Opistorchis and other trematodes that promote cholangiocarcinomas in cats (Meuten, 2020; Tumors in Domestic Animals).

561: The authors state some papers may have been missed by only using pubmed. Does using another database reveal more papers? Has that been checked?

R: At the time of the initial search on Web of Science database, the authors did not find additional relevant articles than Pubmed. From that moment, WS tool has not been used for the present review. If the reviewer deems it relevant, we may carry out a new search, updated to the present date.

576: Sounds like the authors are making a statement about their own work, then cite Wypij et al. We could rephrase this.

R: These two neoplasias share similar clinical features, molecular markers and biological behavior. For this reason, cats with FOSCC have been suggested as a spontaneous animal model for the study of HNSCC (Wypij, 2013).

Abstract: suggest a rewrite after results of manuscript are clarified, it should be more succinct and not overstate significance of findings.

R: The Introduction section was improved according to the reviewer's suggestions.

Response to Reviewer #2

The manuscript "Feline oral squamous cell carcinoma: systematic review of etiologic factors" by Sequeira I. et al., Ref: Manuscript ID: vetsci-1869759, is an excellent systematic review on etiological factors of feline oral squamous cell carcinoma that is the most oral neoplasia in cats, locally invasive and with a high mortality rate. It is obvious that a detailed search of the literature has been made, and the outcome has been presented in detail. However, for it to be published in this journal, I invite the authors to insert the forest plot for random effects meta-analysis of the different etiological factors of feline oral squamous cell carcinoma.

R: We are very grateful for your comments about our work.  As mentioned in the manuscript, this study does not meet the requirements for a quantitative systematic review. The authors consider that we can define this review as a critical review. We emphasize that it was not possible to carry out a statistical analysis, since only a small number of studies included values or statistical data that could be subjected to quantitative systematic analysis. This is a limitation for meta-analysis and thus making it impossible to insert the forest plot for random effects.

Several changes were made that surely led to improvement of the quality of the manuscript. They are shown in the marked version using the track changes tool.

Reviewer 2 Report

The manuscript "Feline oral squamous cell carcinoma: systematic review of etiologic factors" by Sequeira I. et al., Ref: Manuscript ID: vetsci-1869759, is an excellent systematic review on etiological factors of feline oral squamous cell carcinoma that is the most oral neoplasia in cats, locally invasive and with a high mortality rate. It is obvious that a detailed search of the literature has been made, and the outcome has been presented in detail. However, for it to be published in this journal, I invite the authors to insert the forest plot for random effects meta-analysis of the different etiological factors of feline oral squamous cell carcinoma.

Author Response

We are very grateful for your comments about our work.  As mentioned in the manuscript, this study does not meet the requirements for a quantitative systematic review. The authors consider that we can define this review as a critical review. We emphasize that it was not possible to carry out a statistical analysis since only a small number of studies included values or statistical data that could be subjected to quantitative systematic analysis. This is a limitation of meta-analysis and thus makes it impossible to insert the forest plot for random effects.

Several changes were made that surely led to an improvement in the quality of the manuscript. They are shown in the marked version using the track changes tool.

Reviewer 3 Report

Dear all, this paper is very important to small animal oncology and brings new insights.

Author Response

We are very grateful for your comment. The authors made changes with the aim of further improving the quality of the manuscript. Changes are shown in the marked version of the manuscript using the track changes tool. 

Round 2

Reviewer 2 Report

The manuscript "Feline oral squamous cell carcinoma: systematic review of etiologic factors" by Sequeira I. et al., Ref: Manuscript ID: vetsci-1869759, is an excellent systematic review on etiological factors of feline oral squamous cell carcinoma that is the most oral neoplasia in cats, locally invasive and with a high mortality rate. It is obvious that a detailed search of the literature has been made, and the outcome has been presented in detail. However, for it to be published in this journal, I invite the authors to insert the forest plot for random effects meta-analysis of the different etiological factors of feline oral squamous cell carcinoma.

Author Response

As mentioned in the manuscript, this study does not meet the requirements for a meta-analysis. This is considered a critical review. The forest plot graphically represents the set of effect size together with the confidence intervals of each one of them resulting from the included studies. However, for the forest plot to be generated, the studies must present parameters such as means, variances, samples, p-values or test results that can be included in the meta-analysis. For that reason, it was not possible to carry out a meta-analysis, since the selected studies did not include statistical data that could be subjected to the referred analysis. Changes were made to improve the quality of the manuscript are marked using the track changes tool.